# Potential of Anti-CMV Immunoglobulin Cytotect CP^®^ In Vitro and Ex Vivo in a First-Trimester Placenta Model

**DOI:** 10.3390/microorganisms10040694

**Published:** 2022-03-23

**Authors:** Perrine Coste Mazeau, Chloé Jacquet, Clotilde Muller, Mathis Courant, Chahrazed El Hamel, Thierry Chianea, Sébastien Hantz, Sophie Alain

**Affiliations:** 1RESINFIT, UMR1092, University of Limoges, 2 Rue du Pr Descottes, 87000 Limoges, France; chloe.jacquet@umu.se (C.J.); clotilde.muller@unilim.fr (C.M.); sebastien.hantz@unilim.fr (S.H.); 2National Institute of Health and Medical Research INSERM, UMR 1092, 2 Rue du Pr Descottes, 87000 Limoges, France; mathis.courant@unilim.fr; 3National Reference Center for Herpesviruses, Virology Department, CHU Limoges, 2 Rue Martin Luther King, 87000 Limoges, France; 4Gynecology and Obstetrics Department, CHU Limoges, 87000 Limoges, France; 5Mother and Child Biobank (CB-HME), Pediatric Department, Hôpital de la Mère et de l’Enfant, CHU Limoges, 87000 Limoges, France; chahrazed.elhamel-belili@chu-limoges.fr; 6Department of Biochemistry and Molecular Genetics, CHU Limoges, 87000 Limoges, France; thierry.chianea@chu-limoges.fr

**Keywords:** congenital cytomegalovirus, hyperimmune immunoglobulins, Cytotect CP^®^

## Abstract

Background: Congenital CMV infection is the leading cause of neonatal neurological deficit. We herein studied in vitro and ex vivo the potential of the hyperimmune globulin Cytotect CP^®^ (Biotest, Germany) for congenital infection prevention and treatment. Methods: In vitro neutralization assays were conducted in fibroblasts and retinal epithelial cells on the CMV strains TB40/E and VHL/E to determine the 50% and 90% neutralizing doses (ND50 and ND90). The toxicity was assessed by measuring LDH release. Ex vivo assays were conducted in first-trimester villi explants with the TB40/E strain, namely, neutralization assays, the prevention of villi infection, and the inhibition of viral replication in infected villi. Viability was assessed by β-HCG quantification in supernatants. Results: The in vitro neutralization tests showed that Cytotect CP^®®^ inhibits the development of infection foci (DN50: 0.011–0.014 U/mL for VHL/E and 0.032–0.033 U/mL for TB40E) without any toxicity. In the ex vivo neutralization assays, the DN50 were 0.011 U/mL on day 7 and 0.093 U/mL on day 14. For the prevention of villi infection, the EC50 was 0.024 U/mL on day 7. Cytotect-CP^®^ did not inhibit viral growth in infected villi. No impact on villi viability was observed. Conclusions: These results sustained that Cytotect CP^®^ has the potential to prevent CMV congenital infection.

## 1. Introduction

Humans are the only reservoir of human cytomegalovirus (HCMV), a virus of the *Herpesviridae* family. This virus makes use of multiple excretion routes, including saliva, urine, sexual secretions (semen and cervicovaginal secretions), and breast milk. Pregnancy, due to the induced immunodepression, and childbirth are situations with the risk of maternal infection and also of congenital infection. Even though the mechanism of CMV infection in utero is not entirely understood, some placental tissues, such as amniotic membrane, decidua, and villi, are known to support viral replication [1,2], and viral particles have been detected in epithelia of decidua endothelial glands and also in floating villi [3,4,5].

Congenital CMV infection (cCMV) is the leading cause of viral neonatal neurological deficit and non-genetic hearing loss [6,7] and is the most common viral congenital infection, with an estimated birth prevalence of 0.2–6% in industrialized countries. Every year in France, 3400 children are born infected with CMV. The prevalence of primary CMV infection during pregnancy is about 1 to 2% in western Europe and the United States [7,8]. With a risk of 30% of maternal transmission in cases of CMV primary infection, the prevention of viral transmission to the fetus, particularly during the first trimester of pregnancy, is essential to avoid neurological sequelae in newborns.

Treatment during pregnancy may prevent maternal-fetal transmission or, in the case of a fetal infection, may prevent a developmental disorder. Several studies have focused on the efficacy of hyperimmunoglobulin (HIG) to prevent maternal-fetal transmission with disparate results [9,10,11,12,13,14,15,16]. Despite the interest of hyperimmune immunoglobulins in the preventive treatment of transplanted patients [17,18] and the efficacy of the molecule in the prevention of congenital CMV infection in published retrospective studies, anti-CMV antibodies (Cytogam^®^ and Cytotect^®^) have not proven their efficacy in the only two randomized trials published to date on the subject [9,13]. Differences in the recruitment of subjects based on the time of seroconversion, HIG concentration, and frequency of treatment administration determine the efficacy [10,19]. A third phase III assay was recently launched to evaluate higher concentrations in early pregnancy with a new protocol of administration (NCT 05170269). Thus, understanding the HIG mechanism of action remains an important goal. Innate and adaptive immune responses could be mediated by the immunomodulatory effects of anti-CMV immunoglobulin (CMVIG) preparations that would also provide passive immunization [17] and could help to control some of the direct and indirect effects of CMV infection. However, these preparations are not necessarily identical in terms of avidity, neutralizing potential, specific anti-CMV IgG content, or immunomodulation. Although the effects of antibodies on HCMV transmission have been documented before, especially by Tabata et al. in their model of placental infection and prevention of infection by neutralizing monoclonal antibodies [20], the time of administration and the specific effect of HIG (Cytotect^®^) on HCMV infection at the placental level are not fully documented.

We, thus, aimed to characterize the potential of the hyperimmune globulin Cytotect CP^®^ (Biotest, Germany) as a candidate for congenital infection prevention or treatment in our first trimester placenta model.

## 2. Materials and Methods

### 2.1. Cell and Viruses

**Cells:** Human embryonic lung fibroblast (HEF) (MRC-5 cells, bioMérieux, Craponne, France) cells and epithelial cells (ARPE-19, ATCC^®^ CRL-2302, Molsheim, France) were cultured in minimum essential medium (MEM, Eurobio scientific, Courtaboeuf, France) and minimal essential medium (DMEM F12, Fisher bioblock, Illkirch, France), respectively, and both were supplemented with 10% fetal bovine serum (FBS) (Eurobio scientific, Courtaboeuf, France), 50 μg/mL penicillin, and 10 μg/mL gentamycin. The cells were seeded into 48-well plates (10^5^ cells/well) and incubated for 5 days at 37 °C in 5% CO_2_ until confluence was reached.

**Viruses**: The HCMV laboratory endotheliotropic strains TB40/E and VHL/E were kindly provided by Stephane Chavanas (UMR 1043, CPTP, Toulouse, France). Cell-free virus stocks of the TB40/E and VHL/E strains were obtained after consecutive passages on a confluent monolayer of ARPE-19 on 25, 75, and 175 cm^2^ culture flasks with cytopathic effects up to 90–100%. After one more passage in a 175 cm^2^ culture flask, supernatant was collected when the cytopathic effect reached 100%, clarified by centrifugation for 10 min at 3500 rpm, then stored at −80 °C. The infectious viral titer was determined by a plaque assay with 10-fold dilution steps on MRC-5 or ARPE19 cells and was expressed in foci-forming units per mL (pfu/mL).

### 2.2. Antiviral Compounds

CMV-HIG preparations are purified hyperimmunoglobulin products derived from pooled adult human plasma that are selected for high titers of antibodies against CMV. The plasma is fractionated by ethanol precipitation of proteins to provide a product suitable for intravenous administration. Cytotect CP^®^ is a sugar-free isotonic low-salt solution that contains 5% Ig (50 mg/mL) of which more than 96% is IgG (with approximately 65% IgG1, 30% IgG2, 3% IgG3, and 2% IgG4). The maximum level of immunoglobulin A (IgA) is 2 mg/mL. The anti-CMV content is 100 U/mL, where a unit is defined by the Paul Ehrlich Institute Standard [21]. The Cytotect CP^®^ formulation replaced an earlier preparation in which viral inactivation treatment with b-propiolactone induced modifications of some amino acids, particularly the highly labile Fc fragment of IgG. Other modifications to the original plasma fractionation and purification processes were also implemented. These changes resulted in improved immunomodulatory properties of Cytotect CP^®^, including greater inhibition of allogeneic T cell proliferation and cytokine production than the original Cytotect CP^®^ product [22,23].

Cytotect CP^®^ at 100 units/mL was provided by Biotest AG (Dreieich, Germany) and stored at 4 °C. For in vitro assays, Cytotect CP^®^ was diluted in cell medium to obtain final concentrations of 0.005 U/mL, 0.015 U/mL, 0.05 U/mL, 0.15 U/mL, and 1.5 U/mL. For ex vivo assays, the final Cytotect CP^®^ concentrations used were 0.015 U/mL, 0.15 U/mL, 1.5 U/mL, and 5 U/mL.

### 2.3. In Vitro Assays

#### 2.3.1. Neutralization Assays

In vitro neutralization assays were performed with the VHL/E and TB40 strains as follows: One volume of cell-free virus stock was mixed with one volume of Cytotect CP^®^ to reach a multiplicity of infection (MOI) of 0.1 and Cytotect CP^®^ concentrations of 0.005 U/mL, 0.015 U/mL, 0.05 U/mL, 0.15 U/mL, and 1.5 U/mL in a 450 µL final volume and incubated 1 h at 37 °C before supplementation with 50 µL of FBS in every well. The dilutions were prepared in cell-corresponding medium without FBS. Cell-free virus and Cytotect CP^®^ were then incubated on a cell monolayer of ARPE-19 or HEF in a 48-well plate for 3 h at 37 °C in 5% CO_2_ before renewing the medium. After 5 days of incubation at 37 °C, cells were fixed with 90% acetone (20 min at −20 °C) and stained. Cells were incubated with mouse monoclonal antibodies against immediate early and early HCMV antigens (IEA/EA) (Argène, bioMérieux, Craponne, France) for 30 min at 37 °C at 1:50 in PBS without calcium and magnesium, washed twice for 10 min in PBS, incubated for 30 min at 37 °C with horseradish peroxidase anti-mouse secondary antibody (Dako, Santa Clara, CA, USA) at 1:100 in PBS, and washed twice again. After the addition of 3,3′-Diaminobenzidine (DAB, Dako, Santa Clara, CA, USA), foci of infection were counted with an inverted microscope to determine the 50% and 90% neutralizing doses (ND50 and ND90).

#### 2.3.2. Cytotoxicity Assays

To determine the Cytotect CP^®^ cytotoxic concentrations of 50% and 90% (CC50 and CC90), 50% or 90% cell death was evaluated in 96-well plates on HEF or ARPE-19 cells by measuring LDH release in the cell culture supernatant using a CytoTox 96^®^ Non-Radioactive Cytotoxicity Assay kit (Promega, Charbonnieres Les Bains, France) according to the manufacturer’s instructions in a range from 0.005 U/mL to 0.015 U/mL, 0.05 U/mL, 0.15 U/mL, and 1.5 U/mL of Cytotect CP^®^.

### 2.4. Placental Villi Explants

Placentae were collected from voluntary pregnancy terminations (8–14 weeks of gestation) from HCMV-seronegative women, after consent, in collaboration with the biological resources center of the Mother and Child Hospital of Limoges, France (Collection Biologique HME, CB-HME/CRBioLim, Limoges, France). The HCMV IgG status of the mother was determined by chemiluminescent enzymatic assay on a Liaison^®^ XL analyzer (DiaSorin, Saluggia, Italy). Floating villi (5 mm^3^) biopsies were extracted from the placenta, washed with 0.9% saline solution, and cultured in HEF cell medium.

### 2.5. Ex Vivo Assays

The ex vivo model was adapted from Morère et al. [24] and was previously described in Jacquet et al. [25]. For all ex vivo assays, HEF cells were seeded into 48-well plates at 10^5^ cells/well and then incubated for 5 days at 37 °C in 5% CO_2_ until confluence was reached. HEF infection was performed with the endothelial strain TB40/E. We set up three different protocols to evaluate the efficacy of Cytotect CP^®^ to reflect the reality of congenital infection: (1) neutralization assay before infection, (2) prevention of villi infection by the addition of Cytotect CP^®^, and (3) treatment of already infected villi. For all the experiments, Cytotect CP^®^ was used at the following final concentrations: 0.015 U/mL, 0.15 U/mL, 1.5 U/mL, and 5 U/mL with a viral concentration at an MOI of 1.

**(1) Neutralization assays**. Cell-free virus stock was mixed with Cytotect CP^®^ with the same protocol as the in vitro neutralization assays. The mix was then plated on cells for a 3-h incubation at 37 °C. After medium renewal, the plates were incubated at 37 °C in 5% CO_2_. After five days, sponges (Spongostan dental™, NewPharma, Liège, Belgium) with placental villi explant were added to each well (Figure 1A) and were incubated with the different concentrations of Cytotect CP^®^ at 37 °C in 5% CO_2_.

For the next two protocols, HEF cells were infected with cell-free virus stock of the TB40/E strain at an MOI of 1. The plates were centrifuged for 45 min at 3500 rpm at 37 °C before medium renewal and incubation for 5 days at 37 °C in 5% CO_2_.

**(2) To prevent villi infection** from the HEF HCMV-infected cells, the different concentrations of Cytotect CP^®^ in complete fresh medium were added simultaneously with sponges and villi explants on the infected HEF cells (Figure 1B). Complete fresh medium without the drug was set as a control. The plates were incubated at 37 °C in 5% CO_2_ until the collection of the villi on days 7 and 14.

**(3) For the treatment of already infected villi**, sponges with villi explant were added to infected HEF cells and were incubated at 37 °C in 5% CO_2_. After five days of infection, the sponges and explants were transferred on new plates without cell monolayers (Figure 1C). Complete fresh medium containing the different Cytotect CP^®^ concentrations was added to each test condition.

For each ex vivo condition, villi explants were collected at different time points (days 7 and 14 post-infection or post-treatment for the efficacy assay, meaning day 14 and day 21 post-infection), washed in 0.9% saline solution, and stored at −80 °C. Supernatants were stored for beta-HCG quantification to evaluate placental viability. Assays were performed in triplicate for each condition and on three different placentae. Villi with complete fresh medium without the drug were set as a positive control of infection. Villi with complete fresh medium without the drug or viruses were set as a negative control.

#### 2.5.1. Viral Load in Villi Explants

To quantify viral replication kinetics, total DNA was extracted from samples for each ex vivo condition and the viral load was measured by duplex quantitative PCR.

**Total DNA extraction from villi explants**: Explants were lysed with a 500 mL mix containing 10% proteinase K (Qiagen, Hilden, Germany), 6% Tris, 3% sodium dodecyl sulfate and 81% DNA/RNA-free water at 56 °C until total lysis (20 min). Then, total DNA extraction was performed using NucliSENS^®^ technology on an EasyMag instrument (bioMérieux, Marcy-l’Etoile, France) following the protocol “specific B”.

**A previously described duplex quantitative PCR** for targeting HCMV *UL83* and cell albumin genes was performed to quantify the viral load in villi (CMV copies/10^6^ cells): Total DNA extract (5 μL of each) was mixed with 1× of Perfecta Multiplex qPCR toughmix (VWR International, Fontenay-sous-Bois, France, each primer (250 nM), each probe (200 nM), and DNA/RNA free water to reach a final volume of 25 μL. The qPCR cycling protocol was: 30 s at 95 °C, then 45 cycles of 45 s at 60 °C on the CFX96 Touch Real-Time PCR Detection System (Bio-Rad laboratories). The HCMV and albumin copies were quantified using a range of plasmids with the *UL83* or albumin gene using CFX Manager Software (Bio-Rad laboratories, Hercules, CA, USA). The primer sequences were: *UL83* (5′-GTCAGCGTTCGTGTTT CCCA-3′ and 5′-GGGACACAACACCGTAAAGC-3′) and albumin (5′-GCTGT CATCTCTTGTGGGCTGT-3′ and 5-AAACTCATGGGAGCTGCTGGT T-3′). The probes were: HCMV (Cyanine 5)- CCCGCAACCCGCAACCCTTGATG- (BHQ3) 3′ and albumin (6FAM)- CCTGTCATGCCCACACAAATCTCTCC- (TAMRA) (Eurogentec, Seraing, Belgium) [25].

#### 2.5.2. β-hCG Dosage for Cytotoxicity in Villi Explants

Explant viability was assessed on days 7 and 14 in the previously mentioned tests. Supernatant was aliquoted and stored at −80 °C for β-hCG concentration measurement. β-hCG concentration was measured by an electroluminescent “sandwich” ELISA on magnetic microparticles (Cobas, Roche, France). The results were expressed in mUI per 10^6^ cells (using albumin coding sequence quantification as a reference). The concentration of hyperimmune globulins was considered toxic when β-hCG was significantly higher than the positive control villi without Cytotect CP^®^.

### 2.6. Statistical Analyses

Statistical analyses were performed in GraphPad Prism software (GraphPad Software, San Diego, CA, USA) by a dose-response analysis. In the ex vivo assays, the ND50 or EC50 were calculated with the “IC 50 calculator” software from AAT Bioquest and with GraphPad Prism software. Comparison analyses were performed using two-way ANOVA tests and Mann–Whitney tests for non-parametric data in GraphPad Prism software. Statistically significant differences were defined by a *p* value lower than 0.05.

## 3. Results

### 3.1. In Vitro Assays

The results of the neutralization assays on fibroblast and epithelial ARPE-19 cells performed against the VHL/E and TB40/E strains are presented in Figure 2. The ability of Cytotect CP^®^ to neutralize infection by the HCMV strain showed similar profiles on both cell types. The ND50 and ND90 values were higher for the TB40/E strains (Table 1), although the difference with the VHL/E strain was not significant (*p* > 0.9). Cytotect CP^®^ neutralizes HCMV infection of fibroblasts and epithelial cells in vitro.

Cytotect CP^®^ showed no cytotoxic effect on HEF and ARPE-19 cells, even for the highest concentration tested (20 U/mL). The CC50 and CC90 values were, therefore, not reached (data not shown).

### 3.2. Ex Vivo Assays

#### 3.2.1. Replication of TB40/E in Villi Explants

Before efficacy analysis, the kinetics of TB40/E replication in the villi explants were checked from the positive control of infection (villi with viruses and no immunoglobulins) to assess the reliability of the ex vivo assays. The mean viral load in the villi was measured on days 7, 14, and 21 post-infection in the positive controls from all assays (respectively, six, nine, and three placentae). The mean viral load was 28,136 copies/10^6^ ± 20,429 on day 7, 166,018 copies/10^6^ ± 240,632 on day 14, and 536,489 copies/10^6^ cells ± 621,515 on day 21. Although not significant, we observed a regular increase between days 7 and 21 (*p* = 0.07) (Figure 3).

#### 3.2.2. Impact of Cytotect CP^®^ on Viral Replication

(1) Neutralization Assays

When infection was carried out after the neutralization of extracellular virus, the ND50, measured 7 and 14 days post-infection of the villi explants were, respectively, 0.011 UI/mL (95% CI: 0.005–0.025) and 0.093 UI/mL (95% CI: 0.019–0.841) (*p* = 0.1712), a concentration almost 10 times higher on day 14 (Figure 4). Cytotect CP^®^ showed a neutralizing effect, with a significant decrease in the viral load from the positive control of 0.15 UI/mL on day 7 (*p* = 0.029) and 1.5 UI/mL on day 14 (*p* = 0.037).

On day 14, we observed a non-significant increase in the viral load at a Cytotect CP^®^ concentration of 0.015 U/mL.

(2) Prevention of Villi Infection by the Addition of Cytotect CP^®^ at the Time of Explant Infection

On day 7 post-infection, we observed a regular decrease in the viral load, with increasing concentrations but no significant decrease in the viral load in the explant, although the EC50 was reached at 0.024 U/mL (95% CI: 0.0007–0.3237). We did not observe any effect of the molecule on day 14, and we also observed on day 14 a non-significant increase in the viral load at a Cytotect CP^®^ concentration of 0.015 UI/mL (Figure 5).

(3) Treatment of Infected Villi

When Cytotect CP^®^ was added after infection of the villi, the only concentration that showed a decrease in the viral load was 1.5 UI/mL, but EC50 was not reached with increasing concentrations of Cytotect CP^®^ in these efficacy assays. We also observed on day 14 a non-significant increase in the viral load at a Cytotect CP^®^ concentration of 0.015 U/mL (Figure 6).

#### 3.2.3. Placenta Viability

As in the in vitro assays, Cytotect CP^®^ did not show toxicity on placental villi (Figure 7).

On day 7, the mean β-hCG level in the negative control villi was 13,482 ± 22,986 mUI β-hCG/10^6^ cells compared to 14,055 ± 15,978 in the positive control villi without Cytotect CP^®^ (*p* = 0.46). On day 14, the β-hCG level in the negative control villi was 15,904 ± 31,553 mUI β-hCG/10^6^ cells compared to 12,859 ± 13,995 in the positive control villi without Cytotect CP^®^ (*p* = 0.48). On day 21, the β-hCG level in the negative control villi was 5096 ± 5930 mUI β-hCG/10^6^ cells compared to 10,124 ± 10,076 in the positive control villi without Cytotect CP^®^ (*p* = 0.69). There was no significant difference in the β-hCG levels of the native villi (negative control) and the villi infected without immunoglobulin (positive control) (Figure 8A).

We also compared the median β-hCG level in negative control villi and villi with Cytotect CP^®^ at the maximum dose of 5 U/mL used in our ex vivo assays without TB40/E (Figure 8B). We did not find any significant difference on either day 7 (*p* = 0.8148) or day 14 (*p* > 0.9999) of exposure to Cytotect CP^®^.

## 4. Discussion

Amongst the molecules effective against CMV, none is approved, to date, for the prevention of maternofetal CMV transmission. While administered to congenitally infected newborns, Valganciclovir^®^ or Ganciclovir^®^ are not currently approved for use in pregnant women due to their hematotoxicity. Other antivirals used in the management of immunocompromised patients (Foscarnet^®^ and Cidofovir^®^) cannot be used in this context due to their renal toxicity. In 2020, Shahar-Nissan et al. published the only randomized trial to date demonstrating that Valaciclovir^®^, an Acyclovir^®^ prodrug, given at a high dose (8 g/d) would reduce the risk of fetal transmission from 30 to 11% for women with a primary infection during the first trimester of pregnancy or periconceptional period (*p* = 0.027; odds ratio 0.29, 95% CI 0.09–0.90) [26]. No side effects have been reported to date in women or fetuses treated with high doses of Valaciclovir^®^ [26].

In parallel, there is a growing interest in the use of antibodies mimicking or enhancing natural defenses at the placental level, but the results from clinical studies are rather disparate. Despite encouraging results in non-randomized studies [11,14,15,27], the efficacy of HIG has not been confirmed in the only two randomized studies published to date. Revello et al. in 2014 showed a rate of maternal-fetal transmission of 30% in the group of patients treated with HIG vs. 44% in the control group (*p* = 0.13) and observed a non-significant increase in the risk of obstetrical events (13% vs. 2%), such as prematurity (7.6%), preeclampsia (1.9%), and intrauterine growth retardation (3.8%), in the treated group [13]. This trial was probably impaired by the use of insufficient doses of immunoglobulins (100 U/kg) that were not frequent enough (monthly administration) with respect to the 11-day half-life of immunoglobulins and delayed (median administration was 5 weeks after the diagnosis of infection) and by the inclusion of patients with a primary infection in the second trimester [15]. In 2020, another randomized clinical trial began in the United States with the same criteria. Initially designed on the basis of 800 patients to be included, the trial was stopped at an interim analysis at 399 patients because of a lack of treatment efficacy: 22.7% transmission in the immunoglobulin group and 19.4% in the placebo group. Prematurity rates were similar in the two groups (12.2% vs. 8.3%) [9]. Gabrielli et al. histologically analyzed HIG effects on placentas collected for the Revello study and showed no significant differences for all tissue damage between the HIG and placebo groups [28]. Based on the recent pharmacokinetics study indicating that the half-life of HIG is only about 10 days [10,29], a new observational study was undertaken in 149 pregnant women to study the efficacy of Cytotect CP^®^ in pregnant women with a very recent primary infection in the first trimester or during the periconceptional period. Cytotect CP^®^ was administered every 15 days until 18 weeks of amenorrhea, on average only four times, at a dose of 200 IU/kg bodyweight in patients with a primary infection before 14 weeks of amenorrhea. Intravenous treatment was initiated within 3 weeks [19]. In this study, the rate of maternal-fetal transmission was significantly lower than in their 2018 cohort (7.5% in the intervention group vs. 35.2% in the control group; *p* < 0.0001) [10]. In 2020, Seidel et al. also published an observational study using the same doses as Kagan et al. and found a significant decrease in the rate of maternal-fetal transmission of 23.9%, regardless of the term of the pregnancy [15], compared to 39.9% in the control group of the historical cohort of Feldman et al. (*p* = 0.026) [30]. These two recent studies did not find an excess risk of obstetrical events. Kagan et al. suggest a possible link with the discontinuation of immunoglobulins in their study at 20 weeks’ gestation. As a curative treatment, Cytotect CP^®^ has also shown encouraging results for continued use [31]. Last year, Tanimura et al. compared the outcome of symptomatic fetuses receiving immunoglobulin fetal therapy (Ig FT) with or without neonatal treatment (NT) with Valganciclovir^®^ or Ganciclovir^®^ versus symptomatic fetuses receiving only neonatal treatment. The proportion of infants with severe disorders in the fetal therapy group was significantly lower than that in the neonatal therapy group (18.2% vs. 64.3%, *p* < 0.05). This is the first study demonstrating that Ig FT in addition to NT may be more effective in improving the neurological outcomes of newborns with symptomatic cCMV as compared to neonatal treatment only [32]. As a whole, these recent results suggest the capacity of HIG to prevent or eventually treat cCMV.

Due to the heterogeneity of these clinical results using different doses and timing protocols and the obstetrical side effects observed in some studies [13,33], we aimed to specifically assess the efficacy and toxicity of Cytotect CP^®^. Therefore, we herein revisited the effect of Cytotect CP^®^ in our first-trimester HCMV infection model that was previously validated for antiviral efficacy testing [24,25]. We adapted it to reflect the three stages of potential inhibition by HIG. The efficacy and toxicity can be measured, and these ex vivo results can be correlated with previous clinical results. Our model preserves cellular and tissue integrity, unlike in vitro models, which use chemical cleavage methods to isolate placental cells, promoting viral permissiveness [34,35], and our three assay times (especially B and C) reflect the reality of congenital infection. Placental villi from the first trimester were used to reflect first-trimester transmission, which is involved in the most severe sequelae [6,7,36]. This model showed robust and reproducible results for the evaluation of direct and indirect anti-CMV drugs inhibiting cellular components with both reference strains and clinical isolates [25]. For this first evaluation, we used cell-free stocks from one reference strain that was proven to infect placental explants: TB40/E. As described previously by Jacquet et al., the endotheliotropic strain TB40/E replicated with a significantly increased rate between day 4 and day 13 and in the first trimester compared to the third trimester [25]. We also found an increase with a significant disparity between placentas. This is one of the limitations of our study, which can be related to the term of the pregnancy, the delay before the use of the placenta, or the individual susceptibility in connection with the innate immunity of each. This justified repeated experiments with different placentae. To our knowledge, it is the first time that such specific assays including both efficacy and toxicity evaluations on placentae with Cytotect CP^®^ are shown. As there are multiple effects of immunoglobulins on placenta infection, each antibody or HIG preparation may result in different immune stimulations. It is, thus, important to specifically test the efficacy and toxicity of the preparation we intend to use in a clinical trial. Neutralizing antibodies have a varying capacity to interfere in vitro with the HCMV entry routes mediating infection of fibroblasts and epithelial cells. However, both may be required to enter and spread in the placenta to the fetus. The infection of fibroblasts and epithelial cells by HCMV requires glycoprotein complexes composed of gB and gH/gL/gO [37]. Neutralizing antibodies directed against gB or gH epitopes interfere with both fibroblast and epithelial cell infections. Epithelial cell infection is additionally dependent on the envelope pentamer complex composed of gH, gL, UL128, UL130, and UL131A. However, neutralizing antibodies primarily targeting the conformational epitopes of the UL128/130/131A subunits cannot prevent fibroblast entry but are effective in blocking epithelial cell infection [38]. Thus, testing the potential of HIG first in various cell types and in more complex systems in ex vivo placental tissues that mimic the maternofetal interface and placental tissues is more relevant.

We initially confirmed in vitro the neutralizing capacity of Cytotect CP^®^ in both the fibroblasts and epithelial ARPE-19 cell lines on the endothelial strains selected for the assays and its absence of cytotoxicity, even at the highest doses. The DN50 and DN90 values were similar in both cell types. They were below the calculated plasma concentrations of 0.15 U/mL after a 400 U/kg administration for preemptive treatment, but the DN90 were close to or over the expected concentration of 0.03 U/mL under the prophylaxis dosage (100 U/Kg). Thus, the first important result was the absence of cytotoxicity of Cytotect CP^®^, both in vitro and ex vivo, with no alteration of explant viability within a prolonged period of observation of 14 days. This is in agreement with previous results from Kagan’s clinical observations [10,19].

Second, Cytotect CP^®^ demonstrated a strong neutralizing capacity against HCMV infection ex vivo, as the contact of the immunoglobulins with the virus clearly limited the placental infection. With the second protocol that more closely reflected the reality of congenital infection, where viruses and antibodies meet at the placental interface, Cytotect CP^®^ seemed to be effective on D7 but not on D14. The 11-day half-life of immunoglobulins could explain the loss of Cytotect CP^®^ effectiveness on D14. Conversely, Cytotect CP^®^ did not show any efficacy on already infected villi and, therefore, for the curative treatment of congenital infection. In all our placentae on D14, we also observed a non-significant increase in the viral load at the 0.015 U/mL concentration of Cytotect CP^®^. One hypothesis is that a low dose of immunoglobulin may not be sufficient to stimulate immunity and may allow viral replication. The increase of ND50 between D7 and D14 in the neutralization assays and the absence of action of the molecule on D14 in our prevention trials could suggest the need to renew the immunoglobulins every 7 days to maintain their effect as in certain pregnancy therapies (platelet alloimmunization, for example, [39]) where their administration schedule is weekly. The hypothesis is in agreement with previous HIG pharmacokinetics studies [10,29]. Further determination of residual neutralizing titers in the assay supernatants over time could also be implemented to check this hypothesis and then new prevention and efficacy ex vivo assays with the renewal of the immunoglobulins on D7. Another hypothesis to explain our results would be that the viral propagation from cell to cell in the explant is slower than cell-free spread. This could explain the inefficiency of Cytotect CP^®^ when added after the infection of the explant in assay C. In the villi infection prevention assay, antibodies could prevent cell-free spread but not cell-to-cell spread, which would explain an effect of Cytotect CP^®^ on day 7 but not on day 14 due to an excessive viral load in the villi.

Indeed, the mechanism by which natural immunoglobulins act at the placental level is not yet elucidated. Anti-HCMV antibodies could promote NK cell activity during viral infection [40]. In addition, IgG has a protective role towards HCMV infection at the maternal–fetal interface [41]. These IgG can bind to FcRn receptors expressed on the surface of syncytiotrophoblasts. However, their role is also double-edged. Indeed, viral particles bound to these IgG could use the ability of IgG to cross syncytiotrophoblasts to infect the fetus [42]. Moreover, these mechanisms can differ between human monoclonal antibodies (mAbs) or different HIG preparations. Tabata et al., in their study, measured the efficacy of a select subset of human mAbs to HCMV attachment/entry factors, including glycoprotein B (gB) and the pentameric complex gH/gL-pUL128–131, in preventing the infection and spread of a clinical strain in primary placental cells and explants of developing anchoring villi [20]. They studied only the infection neutralization (similar to our assay A) and the efficacy on day 1 post-infection (our assay C) with mAbs against HCMV gB (mAb 3–25, targeting site AD-2), gH/gL (mAb 3–16, gH-specific, targeting pentamer site 7), and the pUL128–131 portion of the pentameric complex (mAbs 1–103, targeting pentamer site 3 and mAbs 2–18, targeting pentamer site 1). They also used Cytogam^®^ (HIG) and Synagis^®^, a human mAb to respiratory syncytial virus, in their model of placental infection [20,43,44]. They concluded that neutralizing mAbs directed to pUL128–131 of the pentamer complex gB and gH/gL perform significantly better than Cytogam^®^ in the neutralization of HCMV infection in primary placental cells and reduce infection in anchoring villus explants of first-trimester placentas. Thus, they suggested that treatment with combinations of mAbs, especially those to pUL128–131 of the pentameric complex, could be significantly more effective than current treatments with HIG to reduce HCMV transmission and congenital infection. Cytogam^®^ is a CMV hyperimmune globulin that is derived from human plasma with high anti-CMV antibody titers, which contains a standardized amount of Ig (5 ± 0.1%) with sucrose, human albumin, and sodium as an excipient. In the USA, Cytogam^®^ (CMVIG CG; CSL Behring, Bern, Switzerland) is licensed for the prophylaxis of CMV disease associated with heart, lung, liver, kidney, and pancreas transplantation, whereas in Europe, Cytotect CP^®^ is licensed for the prophylaxis of CMV infection in patients receiving immunosuppressive therapy, particularly after solid organ or other transplants [45]. Germer and Miescher compared in their respective studies Cytotect CP^®^ and Cytogam^®^ with standard immunoglobulins [22,45]. Germer et al. showed consistently high and comparable CMV-binding avidity and recognition of CMV-specific antigens coupled with a similar neutralizing activity against CMV for both HIG in their neutralization assays on fibroblasts [22]. Miesher et al. observed that Cytotect CP^®^ (50 mg/mL IgG and 100 PEIU/mL) contained a somewhat higher ELISA titer of anti-CMV antibodies, but Cytogam^®^ (50 mg/mL Ig) demonstrated a better CMV-neutralizing activity on fibroblasts [45]. Both authors agree that compared to the standard IVIG, CMV-HIG preparations showed a higher CMV-binding activity and wider recognition of the tested CMV-specific glycoprotein antigens as well as improved neutralizing capacity. Although, based on their components and their properties, Cytogam^®^ and Cytotect CP^®^ preparations are not fully identical.

Our results were obtained from a laboratory virus strain and need to be supplemented by testing a congenital virus strain to approximate the mechanism of congenital CMV infection. Moreover, cell-free virus is classically less likely to spread cell-to-cell than a clinical isolate. Thus, with a congenital virus strain, antibodies might be more effective in neutralizing more virus released from infected cells. It is the first time that the potential of the HIG Cytotect CP^®^ (Biotest, Germany) is specifically studied as a candidate for congenital infection prevention or treatment in this first-trimester placenta villi model, and it seems to us that this study brings interesting therapeutic perspectives. As a whole, our results support the need for a weekly effective back-to-back treatment initiated early before placental infection. This therapeutic strategy could be made possible by the implementation of a systematic screening for CMV, which has already been introduced in some centers [46,47].

## 5. Conclusions

Therapies for HCMV represent a major unmet need. The suggested efficiency of Cytotect CP^®^ in prophylaxis is sustained by our results in vitro and in placental villi, showing good efficacy and a low toxicity in different paths of infection. Additional studies will be conducted to evaluate this molecule as a curative treatment.

## Figures and Tables

**Figure 1 microorganisms-10-00694-f001:**
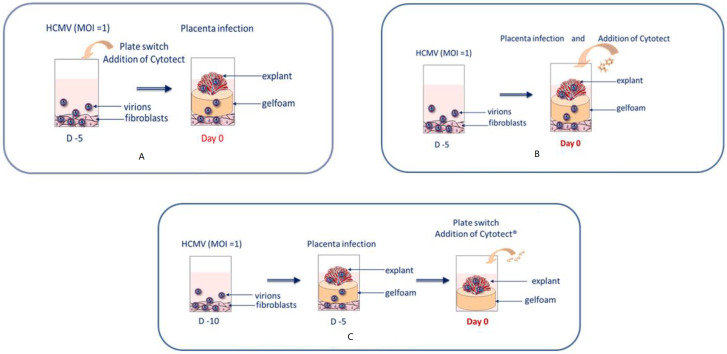
Ex vivo assays. Three different protocols were used to evaluate the efficacy of Cytotect CP^®^. For all ex vivo assays, HEF cells were seeded into 48-well plates at 10^5^ cells/well and then incubated for 5 days at 37 °C in 5% CO_2_ until confluence. The villi explant model is based on infection from cell-free virus produced by infected fibroblasts. HEF cells were infected with cell-free virus stock of the TB40/E strain with a multiplicity of infection (MOI) of 1 in HEF culture medium. The addition of explants on a gelfoam (Spongostan dental™, NewPharma, Belgium) allowed infection by capillarity. We performed three types of assays, mimicking several real-life conditions in maternal blood, at the placental level, and after placental infection (**A**) **Neutralization assay**: The virus was neutralized by Cytotect CP^®^ before the addition of placental explants. TB40/E and Cytotect CP^®^ were incubated on an HEF plate for 3 h at 37 °C in 5% CO_2_ before renewing the medium. After 5 days of incubation, a sponge with a placental villi explant was added in each well and was incubated at 37 °C in 5% CO_2_. Day 0 was the first day of infection of the villi. (**B**) **Prevention of villi infection assay**: Cytotect CP^®^ and explants were added simultaneously after 5 days of infection of the cells. Day 0 was the first day of infection of the villi. (**C**) **Efficacy assay**: Villi were added after 5 days of infection of the cells. After 5 days of infection of the villi, sponges and explants were transferred on new plates without cell monolayers and Cytotect CP^®^ was added. Day 0 was 5 days after placental infection.

**Figure 2 microorganisms-10-00694-f002:**
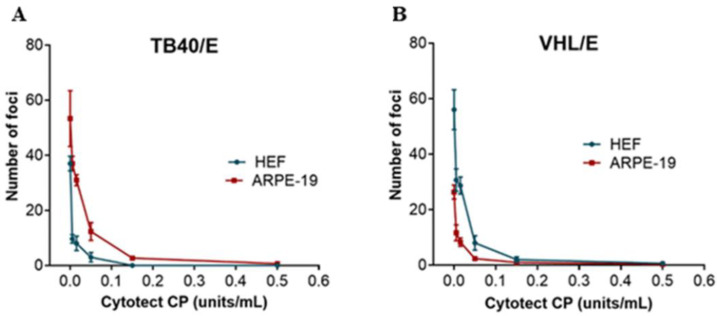
Neutralization activity of Cytotect CP^®^ in vitro. The number of foci after the neutralization assay were counted for (**A**) the TB40/E strain and (**B**) the VHL/E strain on human embryonic fibroblast HEF cells and epithelial ARPE-19 cells after 5 days of incubation (*n* = 3 assays).

**Figure 3 microorganisms-10-00694-f003:**
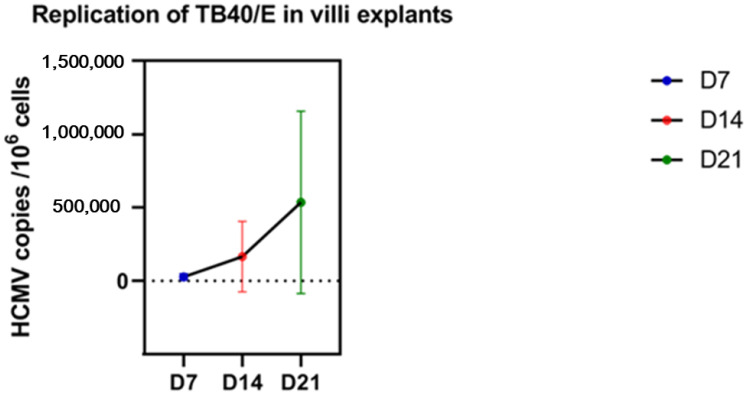
Kinetics of the endotheliotropic strain TB40/E in villi explants. The mean viral loads on days 7, 14, and 21 post-infection in the villi explants. The whiskers are extended to the extreme data points.

**Figure 4 microorganisms-10-00694-f004:**
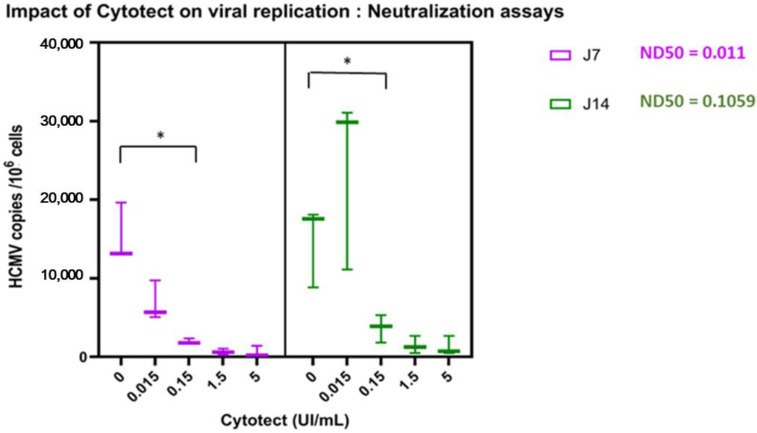
Neutralization assay: ex vivo viral load in villi at 7 and 14 days post-infection. The results were compiled from three placentae. The statistical analyses were carried out using GraphPad Prism with a two-way ANOVA (GraphPad Software, San Diego, CA, USA). * *p* < 0.05.

**Figure 5 microorganisms-10-00694-f005:**
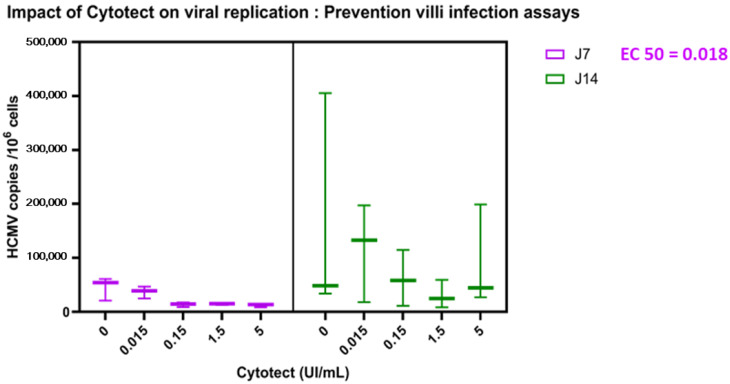
Prevention of villi infection assays. Virus and Cytotect CP^®^ were added at the time of explant infection. The viral loads in the villi were measured at 7 and 14 days post-infection by comparison with the infected and non-treated villi (positive control “0”). The results were compiled from three placentae. The statistical analyses were carried out using GraphPad Prism with a two-way ANOVA (GraphPad Software, San Diego, CA, USA). The whiskers are extended to the extreme data points.

**Figure 6 microorganisms-10-00694-f006:**
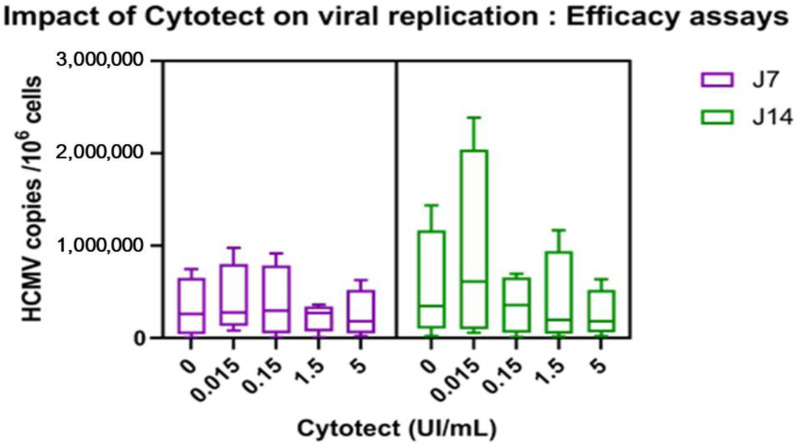
Efficacy assays. Viral loads were measured at 7 and 14 days post-treatment of infected villi and compared with infected and non-treated villi (positive control “0”). The results were compiled from three separate placentae. The boxplots are designed as medians (Q2), lower quartiles (Q1), and upper quartiles (Q3). The whiskers are extended to the extreme data points.

**Figure 7 microorganisms-10-00694-f007:**
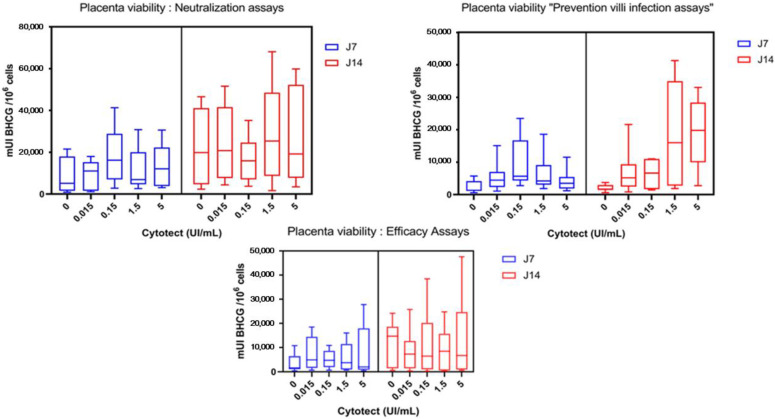
Placenta viability at 7 and 14 days post-infection for the “neutralization” and “prevention” assays and at 14 and 21 days post-infection for the “efficacy” assay. β-hCG units are indicated per 10^6^ cells for each concentration. In this figure, “0” corresponds to the infected villi without Cytotect CP^®^ (positive control). The boxplots are designed as medians (Q2), lower quartiles (Q1), and upper quartiles (Q3) from three placentae. The whiskers are extended to the extreme data points. No significant difference was found between concentrations.

**Figure 8 microorganisms-10-00694-f008:**
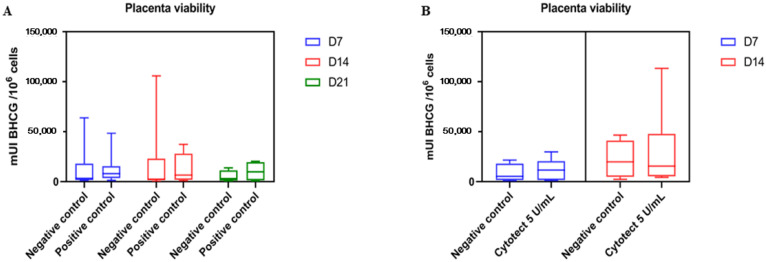
Placenta viability. (**A**) Comparison of β-hCG levels between the negative control villi and villi infected with TB40E without Cytotect CP^®^ after 7, 14, and 21 days of infection. (**B**) Comparison of β-hCG levels from the negative control villi and villi with only Cytotect CP^®^ after 7 and 14 days of exposure to the maximum tested dose. The boxplots are designed as medians (Q2), lower quartiles (Q1), and upper quartiles (Q3). The whiskers are extended to the extreme data points.

**Table 1 microorganisms-10-00694-t001:** In vitro neutralizing efficacy of Cytotect CP^®^. The efficacy of the compound was analyzed against the TB40/E and VHL/E strains on human embryonic fibroblast HEF cells and epithelial ARPE-19 cells. The 50 and 90% neutralizing dose (ND_50_ and ND_90_) values were determined using the results from Figure 2 with GraphPad Prism software.

Viral Strain	HEF		ARPE	
	ND_50_ ± SD (U/mL)	ND_90_ ± SD (U/mL)	ND_50_ ± SD (U/mL)	ND_90_ ± SD (U/mL)
VHL/E	0.014 ± 0.01	0.069 ± 0.02	0.011 ± 0.01	0.067 ± 0.02
TB40/E	0.033 ± 0.01	0.10 ± 0.01	0.032 ± 0.01	0.11 ± 0.02

ND: neutralizing dose; SD: standard deviation.

## Data Availability

The data presented in this study are available on request from the corresponding author.

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
