# Peer review of "Potential of Anti-CMV Immunoglobulin Cytotect CP^®^ In Vitro and Ex Vivo in a First-Trimester Placenta Model"

_microorganisms, 2022, doi:10.3390/microorganisms10040694_

Round 1

Reviewer 1 Report

Mazeu and colleagues have investigated the neutralization potential of anti-CMV immunoglobulins (Cytotect CP) to prevent HCMV infection in a placental culture model. In general, they find that pre-treatment of virus with Cytotect limits infection of the placental culture. In contrast, mixing in the Cytotect antibodies when the placenta is added to the infected culture, had a limited effect and only early after infection, while delaying the Cytotect had no effect on infection of the placenta. In general, the manuscript lacks a meaningful discussion of how this work fits with prior work on anti-CMV antibody products, as well as any discussion of the technical and conceptual limitations of the work. Additionally, the authors raise the hypothesis that anti-CMV antibodies like this might need to be replenished weekly to protect against fetal infection, citing their data that Cytotect  prevented infection of placental explants after 7 days but not 14 days. This concept was interesting, but entirely speculative and could use some additional supporting data. Specific comments are below.

  • The introduction does not really give the reader a picture of the state of the field. Several concerns stuck out for me:
    1. The results of hyper-immune globulin in preventing congenital infection are described as “controversial” in the introduction. This is not really a fair way to introduce the background literature. Small, observational studies have shown mixed results (some with an effect, others without), but these are not random and possibly biased and therefore must be interpreted with study design in mind. Randomized studies (large and small) seem to suggest that hyper-Ig has minimal or no benefit in preventing congenital CMV infection, although there may again be explanations based on study design. The authors address this better in the discussion, but I don’t think it is right to describe these results as “controversial”. I think they need to actually introduce what has been done with CMV Hyper Ig  – perhaps not the pros and cons of each study, but a clear accounting for where the field is now to explain why the current study is needed. Moreover, there are studies of using CMV Hyper Ig in the transplant setting, which might be mentioned somewhere as well.
    2. There is no introduction about Cytotect CP. This product really has to be described for the reader to understand the goals of the study and the authors should not assume that the average reader will know about it.
    3. Similar studies have been conducted with other anti-CMV antibody products (e.g. Cytogam). The introduction could be improved by including prior work with this and other anti-CMV antibody products.
  • The authors would like to argue that the antibody half-life in vitro may explain the lack of effect after 14 days in the "villi infection prevention assay". This is an interesting hypothesis, but it is not supported by the evidence presented. Have the authors attempted to refresh the antibody in the culture after 7 days (or maybe more frequently) as they suggest should be done in clinical trials? Also, it should be possible to determine whether supernatant containing Cytotect and collected after 7, 10 or 14 days retains its neutralization capacity or loses it over time as the authors hypothesize. These studies would improve the manuscript in my opinion, in part because there are alternative explanations besides failure of the Cytotect over time, which are not discussed. For example, it is possible that some of the virus spreads cell-to-cell in the explant, and that this is slower than cell-free spread. In fact, this would explain the complete lack of protection when adding Cytotect after the placenta is infected (the “efficacy assay”). If antibodies fail to block cell-to-cell spread, it might result in a significant change at d7 in the “prevention of villi infection” assay (cell free spread has been blocked), but some villi were infected resulting in slow cell-to-cell spread, eventually resulting in highly infected cultures by day 14. At the minimum, I think alternative interpretations for the results should be discussed.
  • The authors use cell free virus for this study for technical reasons. They should discuss the limitations of this for the assay. It is not simply enough to argue that this model has been validated for testing anti-virals, which this is completely different. Cell-free and highly passaged virus is less likely to spread cell-to-cell than a clinical isolate would. If more of the virus is released from infected cells, it would be more susceptible to antibody neutralization, biasing the results towards protection.
  • Ideally, we might have seen a comparison between Cytotect and other anti-CMV antibody products. However, at the least, the discussion could be improved by discussing Cytotect and the model system. How does this work compare to findings from other antibody products? How does this placental explant model differ and why might it give better results than previous work?

Reviewer 2 Report

The study by Mazeau et al. titled „Potential of anti-CMV Immunoglobulins Cytotect CP® in vitro and ex vivo in first-trimester placenta model“ aims to characterize the potential of hyperimmune globulins Cytotect CP® (Biotest, Germany) to act as a candidate for the prevention of congenital infection. To achieve this goal, the authors have utilized human embryonic fibroblasts, and epithelial cells ARPE-19 and ex vivo assays on human villi explants. The authors show that Cytotect CP® can neutralize infection of cell lines and vili, but does not inhibit virus replication in vili. While this is an important subject of research, the novelty of the data obtained in this study is minor (see below).  

Major comments:

  • The potential of hyperimmune globulins for prevention of congenital CMV infection has been assessed in several clinical studies so far, including Hughes et al NEJM (2021) and Revello et al NEJM (2014). Both studies showed that the administration of hyperimmune globulins did not result in a lower incidence of congenital CMV infection. In addition, many studies have addressed the potential of antibodies to block CMV infection, including the use of hyperimmune globulins to block infection of different cell lines and placenta explants (e.g. Tabata et al Vaccines). Therefore, the results shown in the study by Mazeau et al do not advance the current understanding of the role of antibodies for the prevention of congenital CMV infection.
  • The data on vili infection is rather preliminary and should be supported by additional studies, such as analysis of cell-associated spread of virus in the context of neutralizing antibodies.
  • In general, there are many issues with the manuscript: there are lots of unclear sentences; there is a minimal introduction to new experiments or discussion on obtained results.

Minor comments

  • Figure 1. would better fit together with the results for each experimental approach. As it stands currently, together with minimal description of experiments in the results section and figure legend, it is not easy to follow the experimental approach and results. The manuscript would benefit if models would be explained in more detail, for example, why are human villi explants put on infected HEFs, what is the role of the sponge etc.

The discussion is too long and focuses on various observational studies. Some of the information on observational studies should be included in the introduction. Obtained results should be discussed and put in the context of other studies using the same or similar approaches.  The authors failed to address the lack of significant results and high variability among samples. 

Round 2

Reviewer 1 Report

1) It is not appropriate to quote my review text in the discussion (lines 486-491). Please modify this to be in your own words. I am not an author of the study - taking my words verbatim is not appropriate. This must be changed before this article is published.

2) The authors have now placed this work in the context of the field. However, the novelty and significance of this study is still not substantial. The authors state that no one has investigated this specific preparation of HIG (Cytotect) in this model. However, as pointed out by the authors in the revised manuscript, and in the other review, similar work has been published with other HIG preparations. It would have been better had at least some experiments been done to address the remaining hypotheses and/or to directly compare this product to others.

Author Response

Reviewer 1 :

1) It is not appropriate to quote my review text in the discussion (lines 486-491). Please modify this to be in your own words. I am not an author of the study - taking my words verbatim is not appropriate. This must be changed before this article is published.

We apologize, we have rephrased the sentence as follows : « Another hypothesis to explain our results would be that the viral propagation from cell to cell in the explant is slower than cell-free spread. This could explain the inefficiency of Cytotect CP® when added after infection of the explant in the assay C. In the villi infection prevention assay, antibodies could prevent cell free spread but not cell-to-cell spread which would explain an effect of Cytotect CP® at day 7 but not at day 14 due to too high viral load in the villi. » and « Moreover cell-free virus is classically less likely to spread cell-to-cell than a clinical isolate. Thus, with a congenital virus strain, antibodies might be more effective in neutralizing more virus released from infected cells. »

2) The authors have now placed this work in the context of the field. However, the novelty and significance of this study is still not substantial. The authors state that no one has investigated this specific preparation of HIG (Cytotect) in this model. However, as pointed out by the authors in the revised manuscript, and in the other review, similar work has been published with other HIG preparations. It would have been better had at least some experiments been done to address the remaining hypotheses and/or to directly compare this product to others.

We agree that several very interesting studies have addressed the potential of antibodies to block infection, but the assays we developed specifically addressed the Cytotect CP® preparation, that has not been evaluated this way elsewhere.

As this HIG preparation was used in the recent retrospective studies and in the new beginning phase III study from Kagan’s team, it seemed necessary to focus on it. Morerover it has been suggested by pharmacological studies that the previous phase III studies were not performed in optimal conditions (timing and doses of immune globulins) and thus an insight at the placental level seemed useful to us. Finally, even if both Cytogam® and Cytotect CP® preparations look similar regarding the avidity and neutralization titers, their components differ and their effect on virus spread or villi infection may be different. We believe that our model of placental infection and our three assays times (especially our trial B and C), better reflect the reality of congenital infection than the model of Tabata et al. with a longer time of infection of the villi.

As mentioned in the discussion, we agree that our results and hypotheses need to be confirmed by further assays.

Reviewer 2 Report

While the authors have addressed some of the concerns raised by this reviewer, the overall data are still largely preliminary and should be supported by further experiments as stated in my previous review. In addition, the novelty of the data obtained in this study is minor (also stated in my previous review).  

Author Response

Reviewer 2 :

While the authors have addressed some of the concerns raised by this reviewer, the overall data are still largely preliminary and should be supported by further experiments as stated in my previous review. In addition, the novelty of the data obtained in this study is minor (also stated in my previous review).  

As mentioned in the discussion, we agree that our results and hypotheses need to be confirmed by further assays in future publications.

Round 3

Reviewer 1 Report

The authors have addressed my concerns.

Author Response

Thanks for your constructive comments and approval.